# Arabidopsis O-GlcNAc transferase SEC activates histone methyltransferase ATX1 to regulate flowering

Lijing Xing[1,†], Yan Liu[1,2,†], Shujuan Xu[1,3], Jun Xiao[4], Bo Wang[1,3], Hanwen Deng[1,3], Zhuang Lu[1], Yunyuan Xu[1] & Kang Chong[1,3,5,*] (iD)

## Abstract

Post-translational modification of proteins by O-linked β-N-acetyl-glucosamine (O-GlcNAc) is catalyzed by O-GlcNAc transferases (OGTs). O-GlcNAc modification of proteins regulates multiple important biological processes in metazoans. However, whether protein O-GlcNAcylation is involved in epigenetic processes during plant development is largely unknown. Here, we show that loss of function of SECRET AGENT (SEC), an OGT in Arabidopsis, leads to an early flowering phenotype. This results from reduced histone H3 lysine 4 trimethylation (H3K4me3) of FLOWERING LOCUS C (FLC) locus, which encodes a key negative regulator of flowering. SEC activates ARABIDOPSIS HOMOLOG OF TRITHORAX1 (ATX1), a histone lysine methyltransferase (HKMT), through O-GlcNAc modification to augment ATX1-mediated H3K4me3 histone modification at FLC locus. SEC transfers an O-GlcNAc group on Ser947 of ATX1, which resides in the SET domain, thereby activating ATX1. Taken together, these results uncover a novel post-translational O-GlcNAc modification-mediated mechanism for regulation of HKMT activity and establish the function of O-GlcNAc signaling in epigenetic processes in plants.

**Keywords** ATX1; HKMT activity; O-GlcNAcylation; SEC
**Subject Categories** Chromatin, Epigenetics, Genomics & Functional Genomics; Plant Biology; Post-translational Modifications, Proteolysis & Proteomics
**The EMBO Journal (2018) 37: e98115**

## Introduction

The O-linked β-N-acetylglucosamine (O-GlcNAc) modification of nuclear and cytoplasmic proteins is ubiquitous and essential for a number of biological processes. The dynamic addition and removal of O-GlcNAc at serine and threonine residues are catalyzed by two highly conserved types of enzymes, O-GlcNAc transferase (OGT) and O-GlcNAcase (OGA), respectively (Haltiwanger et al, 1990; Dong & Hart, 1994; Hart et al, 2007). Proteins modified with O-GlcNAc include transcription factors, polymerases, proteasomes, and RNA-processing proteins (Hart et al, 2007), indicating that protein O-GlcNAcylation has significant biological functions. O-GlcNAc modification of core histones has also been identified and is considered as part of the histone code (Sakabe et al, 2010). Recent studies revealed that histone O-GlcNAcylation is coordinated with other post-translational modifications. For example, H2B O-GlcNAcylation at Ser112 facilitates ubiquitination at Lys120 and thus gene transcription (Fujiki et al, 2011). In addition to histones, some proteins associated with chromatin remodeling are modified with O-GlcNAc. In the MCF7 cell line, OGT associates with the enhancer of zeste homolog 2 (EZH2) in the polycomb repressive complex 2 (PRC2), and O-GlcNAc modification of EZH2 at Ser75 plays a role in maintaining EZH2 stability to catalyze H3K27me3 (Chu et al, 2014). O-GlcNAcylation of MLL5β, an isoform of the mammalian trithorax histone lysine methyltransferase (HKMT) MLL5, is critical for recruitment and assembly of the MLL5β-AP-1 transcription activation complex (Nin et al, 2015).

In contrast to the extensive work in animals, the effect of O-GlcNAcylation on epigenetic modulation in plants has been largely unexplored. Unlike for mammalian cells, in which OGT is encoded by a single-copy gene (Kreppel et al, 1997; Shafi et al, 2000), two OGTs, SECRET AGENT (SEC) and SPINDLY (SPY), have been predicted in Arabidopsis (Hartweck et al, 2002, 2006). Recently, it was reported that the DELLA protein RGA is O-GlcNAcylated. O-GlcNAcylation inhibits RGA binding to PHYTOCHROME-INTERACTING FACTOR 3 (PIF3), PIF4, JASMONATE ZIM-domain 1 (JAZ1), and BRASSINA-ZOLE-RESISTANT 1 (BZR1), which in turn coordinately regulate multiple signal pathways in Arabidopsis. The endogenous O-GlcNAc modification of RGA is catalyzed by SEC but not SPY (Zentella et al,

---

1   Key Laboratory of Plant Molecular Physiology, Institute of Botany, Chinese Academy of Sciences, Beijing, China
2   College of Horticulture, Northeast Agricultural University, Harbin, China
3   University of Chinese Academy of Sciences, Beijing, China
4   State Key Laboratory of Plant Cell and Chromosome Engineering, Institute of Genetics and Developmental Biology, Chinese Academy of Sciences, Beijing, China
5   National Center for Plant Gene Research, Beijing, China
    *Corresponding author. Tel: +86 10 62836517; E-mail: chongk@ibcas.ac.cn
    †These authors contributed equally to this work

2016). SPY was later determined to be a novel *O*-fucosyltransferase that catalyzes *O*-fucosylation of DELLA (Zentella *et al*, 2017). In tobacco (*Nicotiana tabacum*), core histones are modified by *O*-GlcNAc and interact with the lectin Nictaba, which is speculated to have a role in chromatin folding (Schouppe *et al*, 2011).

SET-domain proteins function primarily as histone methyltransferases to catalyze histone methylation modification. Most SET proteins function as HKMTs; in addition, some members catalyze methylation of arginine on both histone and non-histone proteins (Ng *et al*, 2007; Niu *et al*, 2007; Wang *et al*, 2007). Trithorax group (TrxG) proteins can catalyze H3K4 methylation. In *Arabidopsis*, five Trithorax homologs (ATX1–ATX5) have been identified and belong to the Trx subfamily of SET-domain proteins based on their conserved SET domain and other characteristic domains; in addition, seven proteins were classified as *ARABIDOPSIS* TRITHORAX-RELATED (ATXR1–ATXR7; Avramova, 2009; Pontvianne *et al*, 2010).

*Arabidopsis FLOWERING LOCUS C* (*FLC*) functions as a central floral repressor (Michaels & Amasino, 1999), and the epigenetic state of *FLC* is critical for regulation of flowering in response to vernalization (Bastow *et al*, 2004; Whittaker & Dean, 2017). *FLC* expression affects the floral transition and is regulated by its active or repressive histone modification state (He, 2009). The *atx1* mutant exhibits an early flowering phenotype correlated with reduced *FLC* expression; ATX1 functions as an activator of *FLC* by targeting to *FLC* chromatin to establish an H3K4me3 modification mark (Pien *et al*, 2008). ATXR7 regulates flowering time by transcriptionally activating *FLC* expression, and *atx1-2 atxr7-2* double mutant shows additive effects on the regulation of flowering time and H3K4 methylation at the *FLC* locus (Tamada *et al*, 2009). Histone methylation plays multiple significant roles during plant development (Liu *et al*, 2010; Pontvianne *et al*, 2010; Thorstensen *et al*, 2011); however, the mechanisms regulating the activity of histone methyltransferases remain poorly understood.

Previous work in winter wheat (*Triticum aestivum*) indicated that *O*-GlcNAcylation-dependent interaction between the lectin protein vernalization-related 2 (VER2) and the RNA-binding protein TaGRP2 regulates vernalization response and flowering by releasing TaGRP2-mediated repression of *TaVRN1* (Xiao *et al*, 2014). In this study, we investigated the effect of loss of function of O-GlcNAc transferase SEC on flowering-time regulation using a null mutant, *sec-5*, in *Arabidopsis*. The *sec-5* mutant exhibits an early flowering phenotype with down-regulated expression of *FLC*. Further biochemical and genetic studies indicated that the histone methyltransferase ATX1 is activated in an *O*-GlcNAcylation-dependent manner. Together, our results not only identify the role of the *O*-GlcNAc transferase SEC in the epigenetic mediation of floral transition but also reveal a novel mechanism regulating HKMT activity in plants.

## Results

### Loss of SEC function results in early flowering and reduced *FLC* expression

To explore whether the *O*-GlcNAc transferase SEC is involved in flowering-time regulation in *Arabidopsis*, we identified two homozygous mutant lines in the Col-0 background, *sec-4* and *sec-5*, which

contain T-DNA insertions in the promoter region and in exon 2 of *SEC*, respectively (Fig 1A and B). Quantitative real-time PCR (qRT–PCR) analysis to detect a transcript between exon 18 and 19 of *SEC*

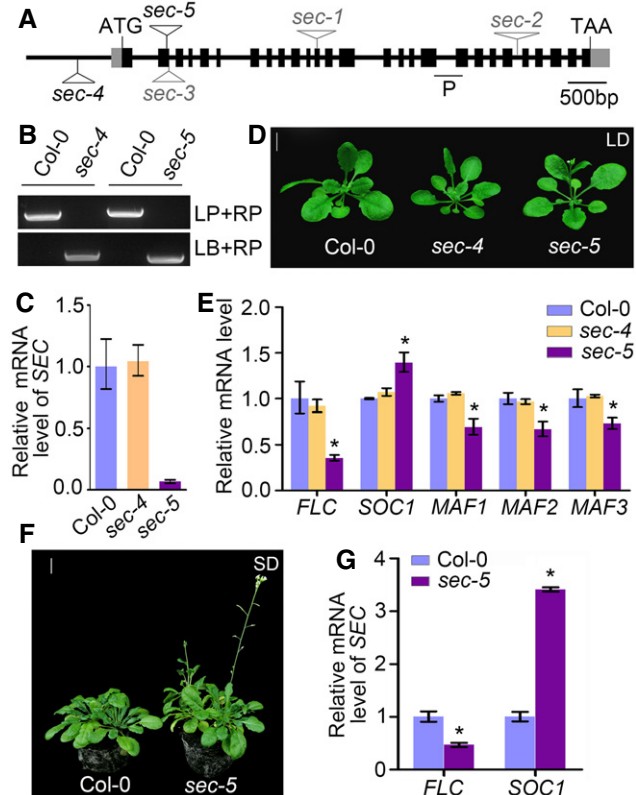

**Figure 1. Identification and characterization of *sec* mutants.**

A  Schematic genomic structure of the *SEC* locus. Exons are shown as black boxes, promoter and introns are indicated as lines, and 5′ and 3′ untranslated regions are shown as gray boxes. T-DNA insertion positions are indicated by triangles. P represents the *SEC* transcript detected by qRT–PCR.

B  Identification of *sec* homozygotes. LB represents the left border primer of the T-DNA insertion. LP and RP represent the left and right genomic primers, respectively.

C  qRT–PCR analysis of *SEC* mRNA levels in *sec-4* and *sec-5* mutants. The expression level was normalized to that of *TUBULIN*, a reference gene for qRT–PCR. Independent biological experiments were repeated three times, and one representative result is shown here. Data shown are means ± standard deviation (s.d.), n = 3.

D  The early flowering phenotype of *sec-5* mutant under long-day (LD) conditions. Scale bar: 1 cm.

E  qRT–PCR analysis of the expression levels of *FLC*, *SOC1*, and *MAF1-3* in 12-d-old vegetative Col-0, *sec-4*, and *sec-5* plants. Independent biological experiments were repeated three times, and one representative result is shown here. The expression level was normalized to that of *TUBULIN*. Data are means ± s.d., n = 3. Statistical significance (two-tailed *t*-test) with *P < 0.05.

F  The *sec-5* mutant exhibits an early flowering phenotype under short-day (SD) conditions. Scale bar: 1 cm.

G  qRT–PCR analysis of *FLC* and *SOC1* expression in Col-0 and *sec-5* plants under SD conditions. The expression level was normalized to that of *TUBULIN*. Experiments were repeated three times, and one representative result is shown here. Data are mean ± s.d., n = 3. Statistical significance (two-tailed *t*-test) with *P < 0.05.

Source data are available online for this figure.

showed that *SEC* mRNA expression was not altered in the *sec-4* mutant but was dramatically lower in the *sec-5* mutant as compared to the wild type (Fig 1C). Further phenotypic observation showed that the *sec-4* mutant had a similar flowering phenotype to Col-0, while the *sec-5* mutant exhibited a reproducible weak early flowering phenotype with reduced rosette leaf number and flowering time compared with wild type under long-day (LD) condition (Fig 1D, Table 1).

The qRT–PCR analysis showed that the key gene of the autonomous and vernalization pathways (Kim *et al*, 2009), *FLC*, was markedly down-regulated in the *sec-5* mutant. In addition, mRNA expression of the *FLC*-clade genes *MADS AFFECTING FLOWERING 1* (*MAF1*), *MAF2* and *MAF3* were slightly down-regulated, and *SUPPRESSOR OF OVEREXPRESSION OF CONSTANS 1* (*SOC1*) was up-regulated, which is consistent with the flowering phenotype of *sec-5*. Nevertheless, the expression level of these genes in *sec-4* was similar to that in Col-0 plants (Fig 1E). Further mRNA expression analysis showed that four genes flanking *SEC*, *At3g04220*, *At3g04230*, *At3g04250*, and *At3g04260*, were unaltered in *sec-5* (Fig EV1A). Thus, the *sec-5* mutant was used for further functional analysis.

When grown under short-day (SD) conditions, *sec-5* plants showed a more obvious early flowering phenotype. The *sec-5* mutant also exhibited down-regulation of *FLC* and up-regulation of *SOC1* transcript levels compared with wild-type plants (Fig 1F and G, Appendix Table S1). In addition, vernalization treatment under SD conditions accelerated flowering of both *sec-5* and wild-type Col-0 plants, and reduced the difference in rosette leaf numbers between *sec-5* and Col-0 plants (Fig EV1B and C, Appendix Table S1). Further qRT–PCR analysis indicated that the transcript levels of key genes in the photoperiod and autonomous pathways, *CONSTANS* (*CO*), *GIGANTEA* (*GI*), *FPA*, *FLOWERING LOCUS D* (*FLD*), *FY*, and *FLOWERING LOCUS K* (*FLK*), were unaltered in *sec-5* under LD conditions (Fig EV1D).

We generated transgenic plants expressing full-length *SEC* cDNA driven by the endogenous *SEC* promoter in the *sec-5* background. These *pSEC::SEC* transgenic lines showed similar flowering-time phenotypes and *FLC* transcript levels to Col-0 under LD conditions (Fig EV1E and F), indicating that the early flowering phenotype of *sec-5* was genetically complemented by *pSEC::SEC* transformation (Fig EV1G and H). The *sec-1* (in Wassilewskija background) and *sec-2* (in Col-0 background) mutants have previously been reported to have T-DNA insertions in exon 12 and intron 22, respectively (Hartweck *et al*, 2002, 2006). The number of days to flowering was unchanged in *sec-1* and *sec-2* as compared with wild type, but both mutants show reduced total leaf numbers at flowering, which has been suggested to result from a reduced leaf production rate of *sec-1* and *sec-2* mutants. The *sec-1* did not show a flowering phenotype neither in long days nor short days (Hartweck *et al*, 2002, 2006), and this may be an ecotype difference. Another allele, *sec-3* (in Landsberg *erecta* background), has been shown to have reduced gibberellin (GA) response and a dwarf phenotype, but its flowering phenotype was not reported (Zentella *et al*, 2016). Our findings support the possibility that *sec-5* accelerates flowering by regulating *FLC* transcription.

To determine the genetic relationship between *SEC* and *FLC*, we generated *35S::SEC*-overexpressing transgenic plants in C24 and *flc-20* (C24 background), respectively. *35S::SEC* overexpression in C24 resulted in a late-flowering phenotype; however, *flc-20 35S::SEC* plants showed almost the same flowering phenotype as *flc-20*, indicating that *SEC* might be functionally dependent on *FLC* in the same genetic pathway for flowering (Fig EV2A–C).

Taken together, our data demonstrate that *SEC* regulates flowering time in an *FLC*-dependent manner, and endogenous *SEC* plays a negative role in regulating flowering time by activating *FLC* transcription.

## Loss of SEC function alters the histone modification of *FLC* chromatin

To investigate whether histone modifications are involved in the down-regulation of *FLC* in the *sec-5* mutant, we analyzed the H3K4me2, H3K4me3, H3K36me3, and H3K27me3 modification states, which are associated with either active or repressive states of *FLC* chromatin, by chromatin immunoprecipitation (ChIP). Because the regions near the transcription start site play important roles in *FLC* transcription activity (Bastow *et al*, 2004), we assayed three regions corresponding to the *FLC* promoter (region P1), transcription start site (region P2), and intron 1 (region P3; Fig 2A, Appendix Fig S1A) for enrichment of various histone modifications. In *sec-5* plants, as compared with Col-0, the H3K4me2 levels were slightly elevated in P1, P2, and P3 (Fig 2B and Appendix Fig S1B), whereas H3K4me3 levels (associated with active transcription of *FLC*) were dramatically decreased, particularly in P2 (Fig 2C and Appendix Fig S1C). H3K36me3, another chromatin modification associated with active transcription of *FLC* (Yang *et al*, 2014), showed nearly identical levels in *sec-5* and wild-type plants in all three regions (Fig 2D and Appendix Fig S1D). The repressive mark H3K27me3 was increased in all three regions (Fig 2E and Appendix Fig S1E). These data indicate that loss of *SEC* significantly decreases the deposition of the H3K4me3 mark and also increased H3K27me3 levels in *FLC* chromatin. H3K4me3 is involved in *FLC* activation, and thus, the reduced expression of

**Table 1.** The *sec-5* mutant exhibits an early flowering phenotype under LD conditions.

| Genotype | Visible buds (%) | Bolting rates (%) | Days to first flower opening | Rosette leaf no. | *n* |
|---|---|---|---|---|---|
| Col-0 | 22.2 ± 1.7 | 8.3 ± 0.4 | 32.7 ± 1.2 | 11.5 ± 1.0 | 72 |
| *sec-4* | 23.6 ± 1.2 | 9.6 ± 1.7 | 32.3 ± 0.6 | 11.3 ± 1.1 | 72 |
| *sec-5* | 63.9 ± 1.4** | 47.2 ± 1.2** | 29.3 ± 0.6* | 9.4 ± 0.9** | 72 |

LD, long-day conditions.
Percentages of plants with visible buds were scored at 21 days after germination, and bolting rates were calculated at 24 days after germination. Experiments were repeated three times, and the values are the means ± standard deviation (s.d.).
Significant differences at **$P < 0.01$ between Col-0 and mutants, and at *$P < 0.05$ by two-tailed *t*-test. *n*, total numbers of plants used in statistical analysis.

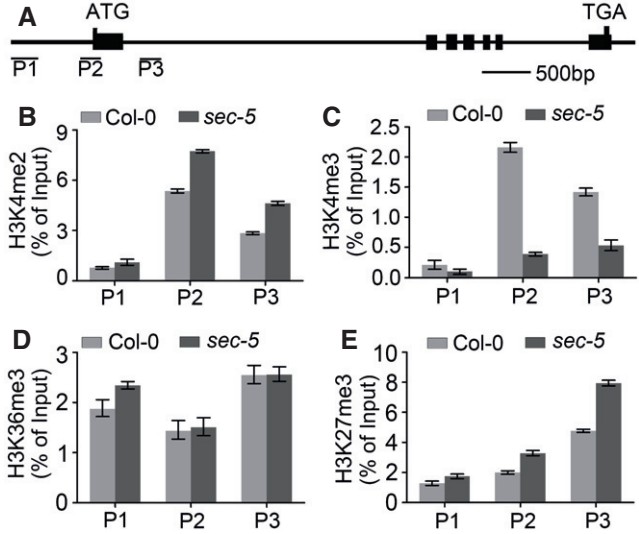

**Figure 2.  Loss of SEC function alters histone modification states on *FLC* chromatin.**

A   Schematic of the *FLC* locus. Exons are indicated with black boxes; promoter and introns are shown as lines. P1, P2, and P3 refer to genomic regions examined by ChIP.
B   ChIP-qPCR assay of H3K4me2 levels of indicated regions at *FLC* chromatin.
C   ChIP-qPCR assay of H3K4me3 levels of indicated regions at *FLC* chromatin.
D   ChIP-qPCR assay of H3K36me3 levels of indicated regions at *FLC* chromatin.
E   ChIP-qPCR assay of H3K27me3 levels of indicated regions at *FLC* chromatin.

Data information: For ChIP analysis, 12-day-old plants were collected and three independent experiments were conducted. Each bar represents the mean ± s.d. of three independent experiments, *n* = 3. The relative abundance was normalized to the input.

*FLC* in *sec-5* correlates with down-regulation of H3K4me3 levels at the *FLC* locus. The data support the proposal that SEC negatively regulates flowering time by maintaining an active *FLC* chromatin state.

### ATX1 interacts with and functionally depends on SEC

ATX1 directly mediates H3K4me3 modification at the *FLC* locus (Pien *et al*, 2008). OGT *O*-GlcNAcylates MLL5β, an isoform of the mammalian TrxG protein MLL5, and acts as a component of the transcription activation complex (Nin *et al*, 2015). This prompted us to explore the possibility that SEC might regulate H3K4me3 modification of the *FLC* locus by catalyzing *O*-GlcNAc modification of the histone lysine methyltransferase ATX1. A yeast two-hybrid assay showed that ATX1 directly interacts with either SEC or SEC-N, a truncated SEC including only the TPR domains at the N-terminus, indicating that these N-terminal TPR domains mediate the interaction between SEC and ATX1 (Fig 3A). To define whether ATX1 is a potential substrate of SEC with O-GlcNAc modification, we expressed HA-tagged ATX1 alone or together with FLAG-tagged SEC in tobacco mesophyll cells. ATX1-HA was immunoprecipitated with anti-HA antibody and then probed with the antibody CTD110.6 which recognizes *O*-GlcNAc sites on serine and threonine residues to detect *O*-GlcNAc modification on ATX1. ATX1-HA co-expressed with FLAG-SEC showed positive *O*-GlcNAc modification, while no signal for the ATX1-HA expressed alone (Fig 3B).

To further understand whether ATX1 is functionally dependent on SEC, we generated *35S::ATX1-HA* overexpression transgenic lines in both the Col-0 and *sec-5* backgrounds. Compared Col-0 plants, homozygous lines overexpressing ATX1-HA in Col-0 displayed a late-flowering phenotype with significantly increased rosette leaf numbers, and dramatically reduced visible buds at 21 days and bolting rate at 24 days after germination. In contrast, *sec-5 35S::ATX1-HA* lines exhibited similar rosette leaf numbers and flowering-time phenotypes as that of *sec-5* (Fig 3C-G). Loss of *SEC* function significantly suppressed the late-flowering phenotype of *35S::ATX1-HA* plants, suggesting that *SEC* is genetically necessary for the function of ATX1 in floral transition.

We also generated a *sec-5 atx1-2* double mutant, and the flowering phenotype analysis showed that *sec-5 atx1-2* exhibited an early flowering phenotype similar to those of *sec-5*, *atx1-2*, and *flc-3* mutants (Fig EV2D–F, Appendix Table S2), indicating that *SEC* and *ATX1* regulate flowering time through the same pathway.

### SEC activates ATX1 by *O*-GlcNAc modification

To explore the regulation of ATX1 activity by SEC, we used the catalytic domains of each protein, which are easier to recombinantly purify than the full-length proteins, for further biochemical analysis. The recombinant ATX1ΔN includes the SET domain (amino acids 899–1017) at the C-terminus, along with the regions immediately upstream (amino acids 801–898) and downstream (amino acids 1,018–1,062) of SET (Appendix Fig S2A; Liu *et al*, 2015). The recombinant SECΔN (amino acids 592–952) contains the complete OGT catalytic domain at the C-terminus of SEC (Appendix Fig S2B). The truncated versions of these two proteins, His-tagged ATX1ΔN and SECΔN, were expressed in *Escherichia coli* and affinity purified (GenScript, Appendix Fig S2C). Biochemical analysis showed that recombinant SECΔN exhibited OGT activity *in vitro* and that *O*-GlcNAcylation of ATX1ΔN could be recognized by the monoclonal antibody CTD110.6. The subsequent β-elimination assay confirmed *O*-GlcNAc modification of ATX1ΔN (Fig 4A). Combined with the down-regulation of H3K4me3 levels at the *FLC* locus in the *sec-5* mutant (Fig 2C), these results suggest that SEC may directly regulate HKMT activity of ATX1 by *O*-GlcNAcylation. ATX1ΔN activity was analyzed using unlabeled *S*-adenosyl-L-methionine (SAM) as the methyl donor. Immunoblot analysis with an antibody against H3K4me3 showed that ATX1ΔN was capable of catalyzing H3K4me3 of recombinant histone H3. When SECΔN and UDP-GlcNAc were added to the reaction system to allow the *O*-GlcNAcylation reaction in advance, the activity of ATX1ΔN was notably enhanced compared with the background activity of ATX1ΔN. In addition, a recombinant mutated H3 (H3K4A) with Lys4 replaced by alanine was not methylated by ATX1ΔN (Fig 4B), indicating that SEC activates ATX1-regulated H3K4 trimethylation.

To investigate further whether SEC *O*-GlcNAcylates ATX1 *in vivo*, we analyzed the effect of loss of SEC function on the *O*-GlcNAcylation level of ATX1 in *Arabidopsis*. The immunoblot assay showed that *O*-GlcNAc modification of ATX1 was detected in Col-0 but not *sec-5* plants, indicating that SEC is necessary for ATX1 *O*-GlcNAcylation *in vivo*. Meanwhile, *sec-5* plants showed reduced ATX1 protein level (Fig 4C). But qRT–PCR analysis showed that loss of *SEC* has no effect on the transcript level of ATX1 (Fig EV2F).

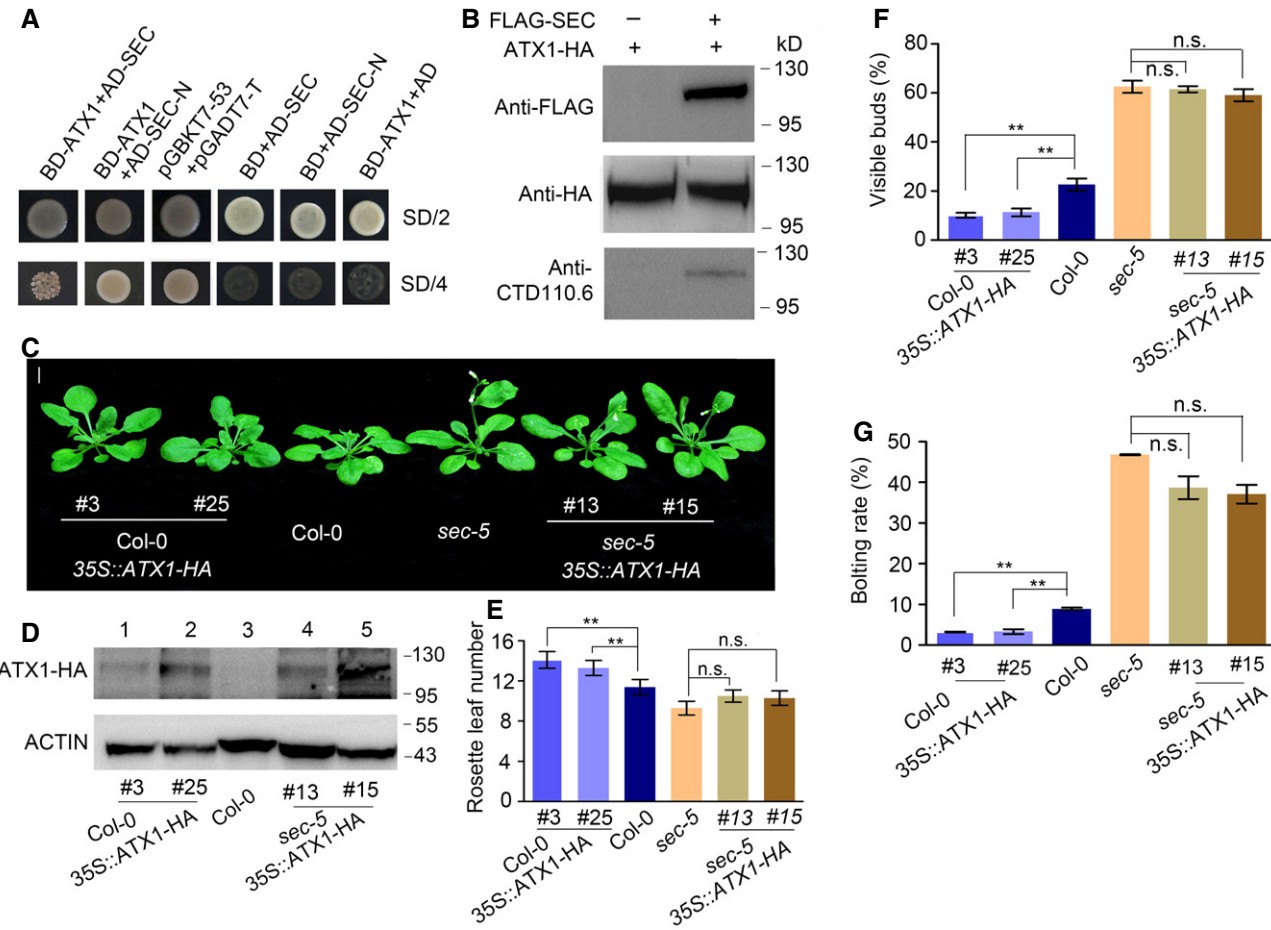

**Figure 3. ATX1 interacts with and is functionally dependent on SEC.**

A    Yeast two-hybrid assay of the interaction between ATX1 and SEC. The full-length coding sequence (CDS) of *ATX1* was cloned into the bait vector pGBKT7. A full-length *SEC* CDS and a truncated fragment were cloned into the prey vector pGADT7 to express complete SEC or truncated SEC-N (including only TPR domains at the N-terminus), respectively. AD vectors expressing SEC or SEC-N were each cotransformed with BD-ATX1. SD/2, SD/-Leu/-Trp medium; SD/4, SD/-Ade/-His/-Leu/-Trp medium. Cotransformed pGBKT7-53/pGADT7-T was used as a positive control.

B    Immunoblot analysis for *O*-GlcNAcylation of ATX1 by SEC using anti-CTD110.6 antibody. HA-tagged ATX1 was expressed alone or together with Flag-tagged SEC in tobacco leaves. Nuclear proteins were extracted from tobacco mesophyll cells, and expression of FLAG-SEC and ATX1-HA was confirmed by Western blotting.

C    Flowering phenotype of *35S::ATX1-HA* overexpression transgenic lines in Col-0 and *sec-5* backgrounds, respectively. Scale bar = 1 cm.

D    Immunoblot analysis using anti-HA antibody to confirm ATX1-HA expression in Col-0 and *35S::ATX1-HA*-overexpressing lines shown in (C). ACTIN was used as a loading control.

E–G    Flowering-time phenotype analysis for Col-0, *sec-5*, and *35S::ATX1-HA* transgenic lines in (C). The percentages of plants with visible buds and the bolting rates were calculated on day 21 and day 24 after plant germination, respectively. Experiments were repeated three times, and for each line, a total of 62 plants were counted for statistical analysis of rosette leaf numbers, visible buds percent, and bolting rates. Data are mean ± s.d., statistical significance (two-tailed *t*-test) with **$P < 0.01$. n.s., not significant.

Source data are available online for this figure.

Subsequently, ATX1 was immunoprecipitated from 12-d-old Col-0 wild-type and *sec-5* plants using anti-ATX1 polyclonal antibody (GenScript) and then used for an HKMT activity assay. As expected, SEC mutation resulted in reduction in the endogenous HKMT activity of ATX1 (Fig 4D). To further investigate the effect of *SEC* mutation on ATX1 protein level, we immunodetected endogenous ATX1 protein in plants at development stages before and after flowering. The data revealed that loss of *SEC* function decreased ATX1 protein level (Fig 4E). Together, these results showed that SEC plays a role in activating the methyltransferase ATX1 through *O*-GlcNAc modification and also in maintaining its stability in *Arabidopsis*.

## S947 is a key site for *O*-GlcNAcylation-dependent activation of ATX1 by SEC

The role of *O*-GlcNAcylation in regulating ATX1 activity prompted us to look for functional *O*-GlcNAc sites in ATX1. The recombinant His-ATX1ΔN was catalyzed by His-SECΔN for *O*-GlcNAcylation modification *in vitro*, and then for higher-energy collisional dissociation (HCD)–mass spectrometry (MS) analysis to identify potential *O*-GlcNAc modification sites. MS analysis identified a peptide in the ATX1 SET domain that has potential sites of *O*-GlcNAc modifications at S947 and/or T953 (Fig EV3). We further investigated the

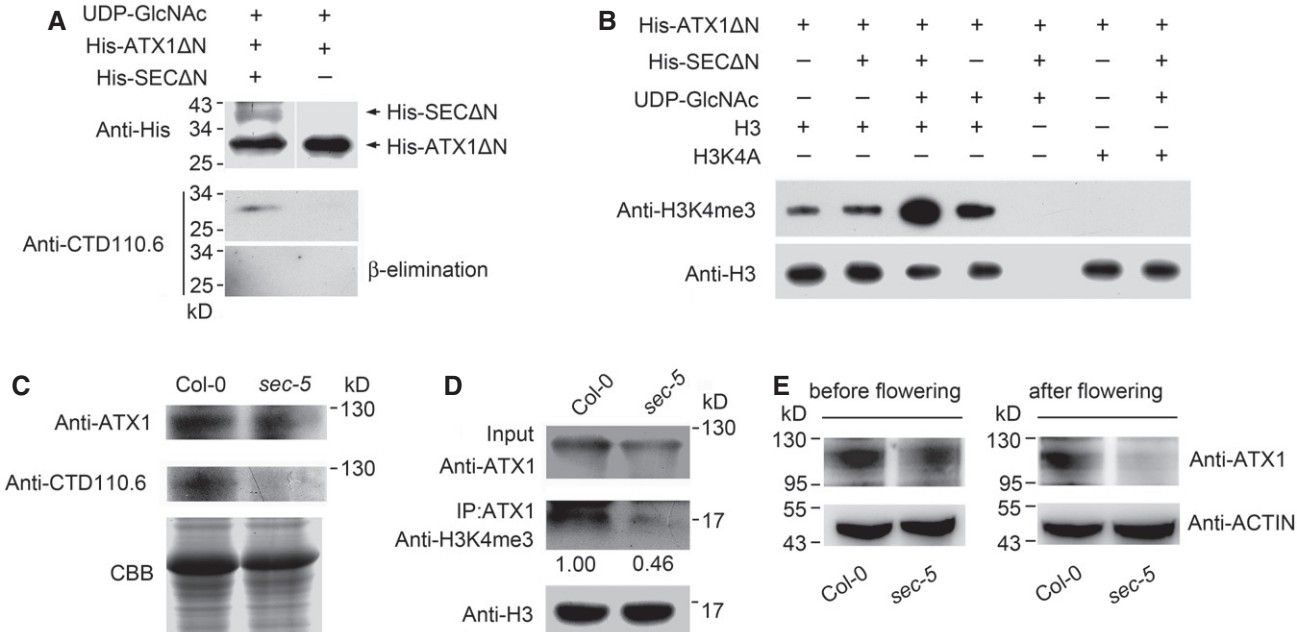

**Figure 4.  SEC *O*-GlcNAcylates and activates ATX1 *in vitro* and *in vivo*.**

A   Detection and conformation of *O*-GlcNAc modification of His-ATX1ΔN catalyzed by His-SECΔN *in vitro*. His-ATX1ΔN (expressing residues 592–952 of the C-terminus) and His-SECΔN (expressing residues 801–1,062 of the C-terminus) were recombinantly expressed and affinity purified separately. *O*-GlcNAcylation of His-ATX1ΔN was detected by anti-CTD110.6 antibody and further conformed by β-elimination analysis.

B   His-SECΔN activates His-ATX1ΔN *in vitro*. Histone methyltransferase activity of ATX1ΔN was detected with or without recombinant SECΔN. H3K4A: mutated H3 in which the fourth amino acid, a lysine (K), was replaced with alanine (A).

C   Analysis of ATX1 *O*-GlcNAc modification in wild-type and *sec-5* plants. Total soluble protein extracts from 12-day-old seedlings were subjected to SDS–PAGE followed by immunoblotting using the indicated antibodies. CBB: Coomassie brilliant blue staining, showing relative protein loading amount.

D   Loss of SEC function reduced ATX1 activity in *Arabidopsis*. Nuclear proteins were extracted, and ATX1 was immunoprecipitated with anti-ATX1 antibody from wild-type and *sec-5* mutant plants, respectively, and then used for histone H3K4 methyltransferase activity analysis with recombinant H3 as catalyzing substrate. Band intensities were quantified with ImageJ. The H3 signal was first normalized by input signal and then was used for H3K4me3 signal normalization.

E   Comparison of ATX1 protein levels in Col-0 and *sec-5* plants at stages before and after flowering. Seedlings were collected for protein extraction at 21 days (before flowering) and 35 days (after flowering) after seeds were planted on plates. Anti-ATX1 antibody was used for immunoblot assay.

Source data are available online for this figure.

possibility of *O*-GlcNAcylation on S947. To explore the potential role of S947 *O*-GlcNAcylation in regulating ATX1 HKMT activity, we mutated S947 to A for recombinant expression of the mutant protein His-ATX1ΔN-m in *E. coli*. Simultaneously, we replaced all 12 serine and threonine residues in the ATX1 SET domain with alanine for expression of another mutant protein, His-ATX1ΔN-12m. Compared with His-ATX1ΔN, His-ATX1ΔN-m and His-ATX1ΔN-12m showed similarly reduced *O*-GlcNAcylation levels (Fig 5A), indicating that S947 is a key O-GlcNAc site in SET domain of ATX1.

Based on the previously reported binary and ternary crystal structure of human OGT (Lazarus *et al*, 2011), we created a model of the overall structure of SEC protein using the Phyre2 web portal (http://www.sbg.bio.ic.ac.uk/phyre2; Kelley *et al*, 2015; Fig EV4A). We found a strong resemblance between SEC and OGT. Compared with human OGT, a small motif corresponding to amino acids 727–929 of human OGT is deleted in the *O*-GlcNAc transferase domain of SEC; however, this motif does not participate in the binding of substrates and peptides to OGT (Lazarus *et al*, 2011). According to our prediction, there are five extremely conserved amino acids, Phe540, His541, His604, Gln776, and Lys779, located within the catalytic region, which are critical for SEC binding with substrates

(Fig EV4B). Among these, His541 and His604 are reported to be conserved and required for SEC activity (Zentella *et al*, 2016). A truncated version of SEC with these five residues replaced with alanine (amino acids 539–952, containing the OGT domain of SEC and 53 amino acids immediately upstream, designated ΔSEC5m) was expressed with His tag in *E. coli* and affinity purified. The *in vitro* biochemical activity assay showed that, compared with His-SECΔN, His-ΔSEC5m activated His-ATX1ΔC relatively weakly. Meanwhile, the single-site mutation (His-ATX1ΔN-m, S947A) and the simultaneous mutation of 12 sites (His-ATX1ΔN-12m) in the SET domain of ATX1 reduced its activation by SEC *in vitro* (Fig 5B).

To further verify the role of His-SECΔN in activating His-ATX1ΔN, we conducted an *in vitro* quantitative HKMT activity assay (EpiGentek). The results showed that His-ATX1ΔN was strongly activated by His-SECΔN and less effectively activated by His-ΔSEC5m. In addition, as compared with His-ATX1ΔN, His-ATX1ΔN-12m and His-ATX1ΔN-m were activated only weakly by His-SECΔN (Fig 5C). These results indicate that the five conserved amino acids are necessary for SEC activity, and the activation of ATX1 by SEC is dependent on *O*-GlcNAcylation of the ATX1 SET domain.

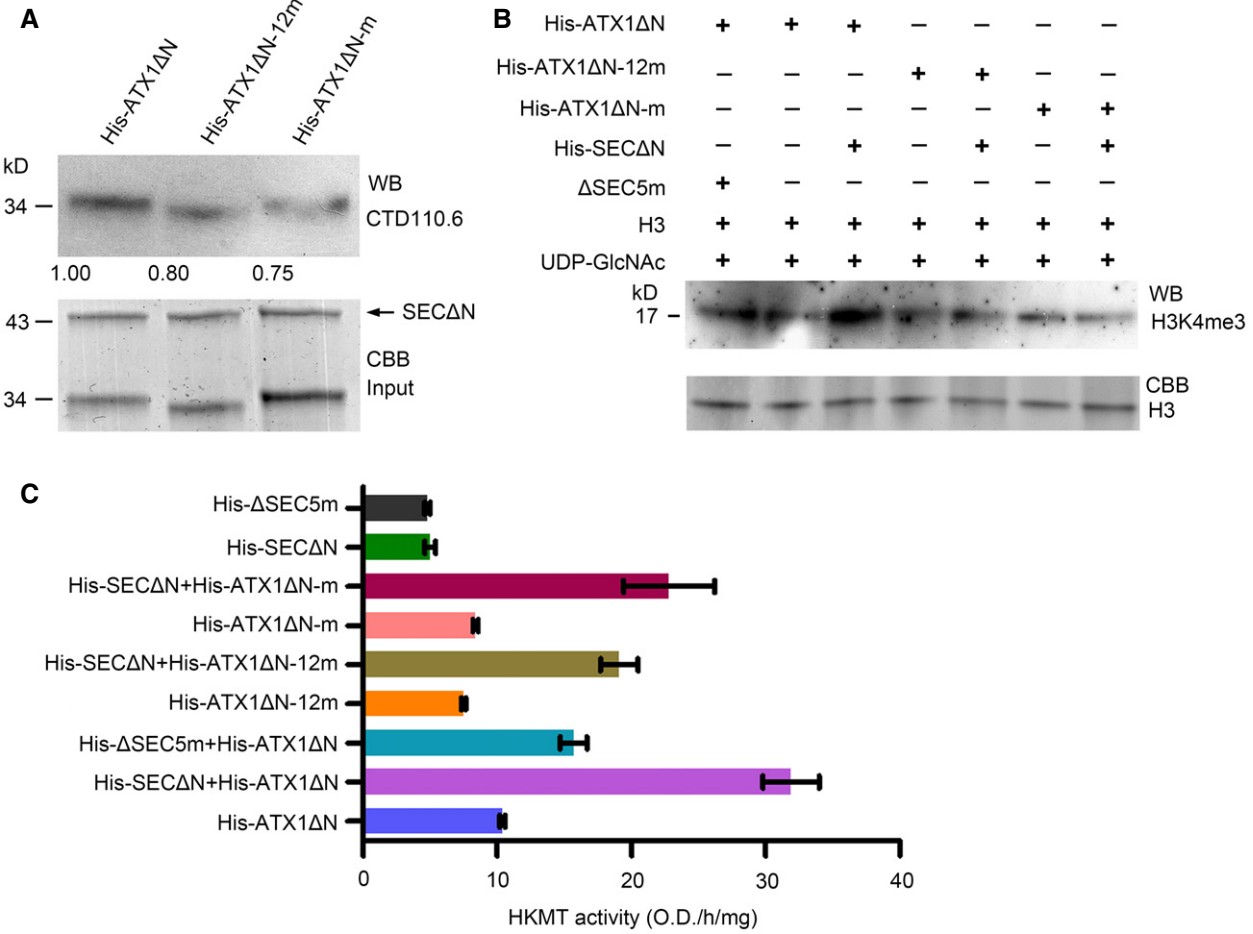

**Figure 5.  S947 of ATX1 is necessary for its *O*-GlcNAc modification and activation by SEC.**

A   Mutation of either S947 alone or all 12 serine and threonine residues in the SET domain reduced *O*-GlcNAc modification level of ATX1ΔN.
B   Site mutation of S947 or all 12 serine and threonine residues in the ATX1 SET domain inhibited the activation of ATX1 by SEC; moreover, mutation of five conserved amino acids in the SEC functional domain inhibited SEC activity.
C   Quantitative analysis of HKMT activity of ATX1ΔN, ATX1ΔN-12 m, and ATX1ΔN-m and the effect of SECΔN and ΔSEC5m in activating those proteins. Experiments were repeated three times. Error bars are s.d. His-ΔSEC5m and His-SECΔN were used as the control.

Source data are available online for this figure.

## S947 on ATX1 genetically mediates the regulation of flowering in *Arabidopsis*

To evaluate the effect of S947 *O*-GlcNAcylation on the biological function of ATX1 in plants, we generated *35S::ATX1-FLAG* and *35S:: ATX1m-FLAG* (in which S947 was replaced with A) transgenic plants in the *atx1-2* mutant background (Fig 6A and B) and assayed their flowering phenotype. The S947A mutation repressed ATX1-activated *FLC* transcription (Fig 6B and Appendix Fig S3). The early flowering phenotype of *atx1-2* was rescued by *ATX1* but not by *ATX1m*, and *ATX1m* transgenic lines flowered earlier than *ATX1* overexpression and Col-0 plants (Fig 6C–E), suggesting that *O*-GlcNAcylation of ATX1 at S947 is required for its function in activating *FLC* to regulate floral transition.

Together, these results strongly suggest that activation of ATX1 is dependent on its *O*-GlcNAcylation in the SET domain, and in particular on S947. In addition, SEC has conserved residues for

substrate binding, which is similar to what is seen in human OGT and important for activating ATX1. These findings support that, in the Col-0 background, SEC acts as a negative regulator to inhibit flowering transition by activating ATX1 and maintaining its stability through *O*-GlcNAcylation, leading to increased expression of *FLC*.

# Discussion

## SEC mediates epigenetic regulation of flowering by catalyzing *O*-GlcNAc modification of HKMT ATX1

The *O*-GlcNAc modification of RNA-binding protein TaGRP2 is known to mediate the vernalization response for flowering in wheat (Xiao *et al*, 2014). In *Arabidopsis* Landsberg *erecta* (L*er*) plants, SEC functions as a positive regulator of GA signaling by modifying the

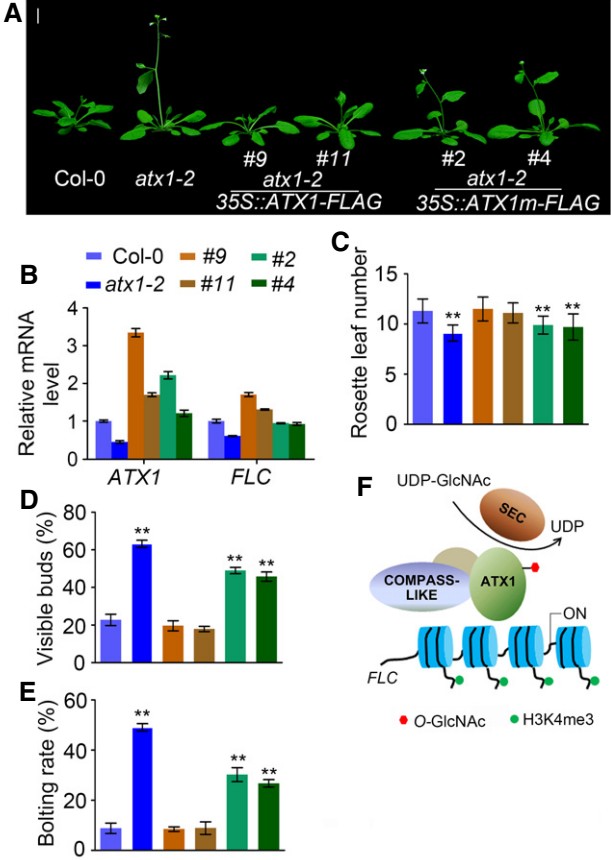

**Figure 6.  Mutation of S947A inhibits ATX1 function in *Arabidopsis*.**

A   Flowering phenotypes of Col-0, *atx1-2*, *atx1-2 35S::ATX1-FLAG*, and *atx1-2 35S::ATX1m FLAG* (with Ser947 replaced with alanine) plants under LD conditions. Scale bar: 1 cm. Two independent lines are shown for each transformation.

B   qRT-PCR analysis of *ATX1* and *FLC* transcript levels in Col-0, *atx1-2*, *atx1-2 35S::ATX1-FLAG*, and *atx1-2 35S::ATX1m FLAG* plants. The transcript levels were normalized to that of *TUBULIN*. Data are mean ± s.d. of three independent biological replicates, *n* = 3.

C–E   Flowering-time phenotype analysis of lines in (A). The percentages of plants with visible buds and the bolting rates were calculated on day 21 and day 24 after plant germination, respectively. Three independent biological repeats were conducted, and data from a total of 71 plants were used for statistical analysis. Data are mean ± s.d., statistical significance (two-tailed *t*-test) with **$P < 0.01$.

F   Working model of SEC activation of ATX1 to regulate *FLC* transcription by *O*-GlcNAc modification.

the level of H3K4me3 in the *FLC* chromatin, leading to repressed *FLC* transcription and an early flowering phenotype. OGT associates mostly with the promoter, transcription start site, and gene body regions to regulate H3K4me3 accumulation (Deplus *et al*, 2013; Vella *et al*, 2013), indicating that *O*-GlcNAc has a role in regulating histone-related gene expression.

In addition to down-regulation of H3K4me3, loss of function of SEC also resulted in an increase in H3K27me3 levels in *FLC* chromatin (Fig 2D). The PRC2 complex maintains a state of transcriptional repression through deposition of H3K27me3 at specific chromatin region, which is conserved in animals and plants (Cao *et al*, 2002; Schubert *et al*, 2005; Whitcomb *et al*, 2007; Xiao *et al*, 2016). In *Drosophila*, the location of *O*-GlcNAc on chromosomes is coincident with Polycomb group (PcG) response elements (PREs; Gambetta *et al*, 2009; Sinclair *et al*, 2009), and PREs and TREs (trithorax response elements) may represent the same regions for regulation of target gene expression (Ringrose & Paro, 2004). The active chromatin mark H3K4me3 specifically inhibits PRC2-mediated H3K27me3 in mammals and flies, and TrxG and PcG proteins antagonistically regulate target gene expression (Papp & Muller, 2006; Schmitges *et al*, 2011). In *Arabidopsis*, PRE-like functional components have been found to regulate H3K27me3 (Berger *et al*, 2011; Xiao *et al*, 2017). During vernalization, the PRC2 complex plays a role in triggering *FLC* silencing in *Arabidopsis* (De Lucia *et al*, 2008). In addition, ATX1 has been suggested to counteract PcG-mediated gene silencing in *Arabidopsis* (Pien & Grossniklaus, 2007). Our data suggest that the increased H3K27me3 observed at *FLC* chromatin in *sec-5* plants might be a result of the down-regulation of H3K4me3. These findings may shed light on *O*-GlcNAc-mediated epigenetic regulation during plant development.

## *O*-GlcNAc modification mediates ATX1 activity and stability regulation in plants

There are five *ATX* genes in the *Arabidopsis* genome, of which *ATX1* and *ATX2* form one subgroup and *ATX3–ATX5* another (Ng *et al*, 2007). As the plant orthologs of MLL proteins, ATX1 and ATX2 have highly similar structures, but they show different temporal and special expression patterns. Despite the high degree of similarity, ATX1 catalyzes H3K4me3 of *FLC* chromatin specifically (Pien *et al*, 2008), but ATX2 mediates H4K4me2 of target genes (Saleh *et al*, 2008). The triple mutant *atx3-1 atx4-1 atx5-1* exhibits dwarf and small rosette leaf phenotypes, but no alteration of flowering time (Chen *et al*, 2017). In contrast to its significant reduction in H3K4me3 in *FLC* chromatin, loss of SEC resulted in slight up-regulation of H3K4me2 in region P3 of *FLC* (Fig 2B), indicating that ATX1 is not required for the deposition of H3K4me2. Non-reduced H3K4me2 has been reported in silenced *FLC* during the vernalization response in *Arabidopsis* (Bastow *et al*, 2004). The slight up-regulation of H3K4me2 levels in *sec-5* mutants suggested that not ATX2 but rather ATX1 activity is influenced by *sec* mutation and that SEC regulates flowering transition by affecting *O*-GlcNAc-dependent ATX1 activity.

*O*-GlcNAc modification plays a role in regulating protein stability and activity (Yi *et al*, 2012; Ruan *et al*, 2013). Tyr1015 is reported to be critical for ATX1 activity, but the regulatory mechanism underlying this effect is unknown (Ding *et al*, 2012). In this study, we presented evidence that SEC *O*-GlcNAcylates ATX1 to activate its

DELLA protein RGA with *O*-GlcNAc to coordinate development signaling pathways, which may be dependent on the genetic background (Hartweck *et al*, 2006; Zentella *et al*, 2016). Here, the *sec-5* mutant showed a wild-type GA response (Fig EV5). Recently, more than 200 proteins with *O*-GlcNAc modification were identified by a large-scale proteomic analysis, suggesting potential roles of *O*-GlcNAc in mediating epigenetic regulation and multiple other cellular processes in *Arabidopsis* (Xu *et al*, 2017). In this study, we provide biochemical and genetic evidence that the *O*-GlcNAc transferase SEC regulates flowering time via an HKMT ATX1-associated epigenetic process in *Arabidopsis*. Loss of function of SEC decreased

histone methyltransferase activity and S947 is critical for *O*-GlcNAc modification-mediated activity regulation of ATX1. According to our results, the S947A mutation did not completely inhibit ATX1 activity (Fig 6B), indicating that there may be other mechanisms for regulation of ATX1 based on HKMT activity. Our results here suggest that *O*-GlcNAcylation acts as a pattern of HKMT activity and stability regulation in plants. It will be of great interest to define whether other TrxG or PcG proteins can also be regulated by *O*-GlcNAc modification in further studies.

### *O*-GlcNAcylation and COMPASS-like complexes

H3K4 methylation is catalyzed by COMPASS (Complex Proteins Associated with Set1) complexes to activate target genes transcription in yeast, human, and plants. The core components of COMPASS complexes are conserved and include Ash2, RbBP5, and WDR5 (Miller *et al*, 2001; Shilatifard, 2008; Ding *et al*, 2012; Xiao *et al*, 2016). In *Arabidopsis*, an Ash2R-containing COMPASS-like complex mediates H3K4me3 in *FLC* to regulate flowering time (Jiang *et al*, 2011). ATX1 acts as one of the components of a COMPASS-like complex to participate in *FLC* activation (Kim & Sung, 2012; Fromm & Avramova, 2014). ATX1 is required for recruitment of AtCOMPASS-like to promoters, but the assembly of the transcription machinery at target promoters is not influenced by the activity of ATX1, indicating that an ATX1/AtCOMPASS-like generated H3K4me3 mark is required for transcription elongation but not initiation (Ding *et al*, 2012).

ATX1-catalyzed H3K4me3 of *FLC* chromatin is required for *FLC* transcription activation. Our data indicate that SEC functions as a negative regulator of flowering to activate ATX1 by *O*-GlcNAc modification, suggesting that *O*-GlcNAc modification-mediated ATX1 activity plays a role in regulating ATX1/AtCOMPASS-like complex function (Fig 6F). In summary, our findings not only uncover a previously unknown mechanism regulating the activity of the TrxG HKMT ATX1, but also establish a distinct epigenetic role of *O*-GlcNAc signaling in plants. OGT is necessary for the formation of the SET1/COMPASS complex. One of the components of the SET1/COMPASS complex, host cell factor 1 (HOS1), is O-GlcNAcylated by OGT to regulate SET1/COMPASS integrity, and OGT activity helps promote SET1 binding to chromatin (Deplus *et al*, 2013). The present study provides novel insight into the regulatory mechanism of the ATX1/AtCOMPASS-like complex in plants. It will be intriguing to explore the role of *O*-GlcNAc modification in regulating HKMT substrate recognition and complex assembly in plants.

# Materials and Methods

### Plant materials and growth conditions

*Arabidopsis thaliana* ecotypes Columbia-0 (Col-0) and C24 were used as wild type and as background for the transgenic lines, respectively. T-DNA insertion mutants of *sec-4* (SALK_106339) and *sec-5* (SALK_034290) were obtained from the SALK T-DNA collection (Alonso *et al*, 2003).

Seeds were sterilized and grown on half-strength Murashige and Skoog medium (1/2 MS) plates for 3 days at 4°C and then transferred to the greenhouse for germination at 22°C with the photoperiods of 16-h light/8-h dark (long day, LD) or 8-h light/16-h dark (short day, SD). Transgenic seeds were grown on 1/2 MS containing 20 mg/l hygromycin as screening pressure to obtain transgenic lines. Flowering time was calculated as rosette leaf numbers at bolting.

### *In vitro O*-GlcNAcylation and methyltransferase assay

1 μg of recombinant expressed His-SECΔN (aa 592–952, containing the OGT domain of SEC) was incubated with 1 μg His-ATX1ΔN (aa 801–1,062, containing the SET domain) and 50 μM UDP-*N*-acetyl-glucosamine in 25 μl of reaction system for 1 h at 37°C. Immunoblotting analysis was conducted to detect *O*-GlcNAc modification of proteins with antibody CTD110.6.

For ATX1 histone methyltransferase enzyme activity assay, purified His-ATX1ΔN was incubated with recombinant expressed H3 (Millipore) and unlabeled *S*-adenosyl-L-methionine (SAM, Sigma) in methyltransferase buffer. After the HMTase assay, the products were analyzed by immunoblot analysis with the indicated antibodies.

### Plasmid construction and plant transformation

To generate a complementation construct for the *sec-5* mutant, a 1,798-bp genomic sequence upstream of the *SEC* ATG and the 2,934-bp cDNA fragment of *SEC* were amplified by PCR and cloned into the vector pFGC5941. To generate the *p35S::SEC* construct, cDNA fragments of *SEC* were amplified by RT–PCR and cloned into the pSN1301 vector. The *p1307-SEC-FLAG* construct was generated by fusion of amplified *SEC* with *FLAG* in the *p1307-cFLAG* plasmid to express the SEC-FLAG fusion protein in plants. To express an ATX1-HA fusion protein in plants, the cDNA fragment of ATX1 was amplified and inserted into the SacI-SalI site of the *pCsVMV-HA3-N-1300* plasmid. The single-site mutation of the *ATX1* cDNA was generated by overlap PCR technique. The mutated products were confirmed by sequencing and cloned to *p1307-cFLAG* plasmid. The primers used for plasmid construction are listed in Appendix Table S3.

The *Arabidopsis* plants were transformed by the floral-dip method (Clough & Bent, 1998) using the GV3101 *Agrobacterium* strain, and positive plants were screened on hygromycin B medium. More than ten independent transformants with a single T-DNA insertion were obtained for each construct. The T3 homozygotes were used for subsequent analysis.

### Reverse transcription PCR and quantitative real-time reverse transcription PCR

Total RNAs were extracted with TRIzol reagent (Invitrogen), and 1 μg of total RNA was used for reverse transcription reaction with SuperScript III reverse transcriptase (Invitrogen). Real-time quantitative reverse transcription PCR (qRT–PCR) analyses were performed using SYBR green mixture (TOYOBO) as described previously (Xiao *et al*, 2014), and *UBIQUITIN* was used as internal control. Three independent biological replicates were performed for each qRT–PCR analysis. The primers used for qRT–PCR are listed in Appendix Table S3.

### ChIP

Chromatin immunoprecipitation (ChIP) experiments were performed according to methods described previously (Bowler *et al*, 2004). Antibodies specific to H3K4me2, H3K4me3, H3K36me3, and H3K27me3 (Millipore) were used to immunoprecipitate *FLC* chromatin. The relative abundances of histone modifications were normalized to input DNA. Primers are listed in Appendix Table S3.

### Yeast two-hybrid assay

For yeast two-hybrid assay, the open reading frame of *ATX1* was amplified and inserted into NcoI and SmaI sites of the bait vector pGBKT7 (BD). The open reading frame of *SEC* was cloned into ClaI and SacI sites of the prey vector pGADT7, and the cDNA fragment of *SEC*-N (cDNA only encoding TPR domains at N-terminus) was cloned into NdeI and EcoRI sites of pGADT7, respectively. Primers used for plasmid construction are shown in Appendix Table S3. To confirm protein interactions in yeast cells, BD-ATX1 was cotransformed with AD-SEC and AD-SEC-N into *Saccharomyces cerevisiae* strain AH109 (Clontech) according to the manufacturer's instructions. Transformants first were grown on SD/-Leu/-Trp (SD/-2) medium, and then, positive clones were screened on SD/-Leu/-Trp/-His/-Ade (SD/4) medium. pGBKT7-53 and pGADT7-T vectors were cotransformed as positive controls, and pGBKT7-Lam and pGADT7-T were cotransformed as a negative control.

### Protein expression, purification, and immunoblot analyses

The cDNA fragments of *SEC* and *ATX1* were cloned into the pET28a vector following codon optimization (GenScript) and used to express peptides corresponding to 361 amino acids of the SEC C-terminus and 262 amino acids of the ATX1 C-terminus, respectively. The same method was used to generate constructs expressing mutated peptides of ATX1-12m (a total of 12 serine and threonine residues were replaced with alanine), ATX1-m (S947 was replaced with alanine), and SEC-5m (Phe540, His541, His604, Gln776, and Lys779 were all replaced with alanine). The recombined plasmids were transformed into the *E. coli* BL21 (DE3) line to generate fusion proteins with His tags. The His-SECΔN, His-ATX1ΔN, His-ATX1-12m, His-ATX1-m, and His-SEC5m fusion proteins were induced and purified by Protein Service (GenScript).

For immunoblot assays, proteins were separated on SDS–PAGE and then electrotransferred to PVDF membranes (Millipore). The membranes were blocked at room temperature for 2 h with TBST buffer (25 mM Tris, 140 mM NaCl, 3 mM KCl, and 0.1% Tween-20, pH 7.4). Membranes were incubated overnight at 4°C with appropriate antibodies and washed with TBST for 3 × 10 min. Then, membranes were incubated with secondary antibody conjugated to horseradish peroxidase (Sigma). Immunoreactive bands were detected by using the enhanced chemiluminescence (ECL) or super ExPlus system.

### In vitro O-GlcNAcylation assay

1 μg of recombinant expressed His-SECΔN was incubated with 1 μg His-ATX1ΔN and 50 μM UDP-*N*-acetylglucosamine in 25 μl of reaction system for 1 h at 37°C. The reaction buffer contained 50 mM Tris–HCl, pH 7.8, 12.5 mM MgCl$_2$, and 1 mM DTT, pH 7.5. Samples were denatured at 95°C for 10 min in 5× loading buffer (100 mM Tris–HCl, pH 6.8, 200 mM DTT, 4% SDS, 20% glycerol, and 0.2% bromophenol blue) and electrophoresed by SDS–PAGE. Immunoblot analysis was conducted to detect *O*-GlcNAc modification of proteins with antibody CTD110.6 specific to *O*-GlcNAc sites.

### β-Elimination assay

The *O*-GlcNAc linkage is easily hydrolyzed in alkaline solution with mild conditions. A β-elimination assay was used to confirm the presence of the *O*-GlcNAc modification on proteins as previously described (Duk *et al*, 1997). The *O*-GlcNAcylated proteins catalyzed by SECΔN were electrotransferred to PVDF membrane (Millipore), and then, membranes were treated with 55 mM NaOH for 16 h at 40°C. The target proteins were redetected with antibody CTD110.6 by immunoblot assay.

### Histone methyltransferase assay

For ATX1 histone methyltransferase enzyme activity assay, 1 μg purified His-ATX1ΔN protein was incubated with 2 μg of recombinant expressed H3 (Millipore) and 1 mM *S*-(5′-adenosyl)-L-methionine iodide (Sigma) in methyltransferase buffer (50 mM Tris, pH 9.0, 1 mM PMSF and 0.5 mM DTT) for 1 h at 30°C. The products were analyzed by immunoblot with anti-H3K4me3 antibody (Millipore). The quantitative analysis of ATX1 histone methyltransferase enzyme activity was based on the amount of methylated H3K4 converted by ATX1 with EpiQuik Histone Methyltransferase Activity/Inhibition Assay Kit (EpiGentek).

### Identification of O-GlcNAc sites by higher-energy collisional dissociation (HCD)–mass spectrometry (MS) and electron transfer dissociation (ETD)-MS

Recombinant His-SECΔN and His-ATX1ΔN proteins expressed in *E. coli* strain BL21 (DE3) were purified (GenScript) and incubated for SEC *O*-GlcNAcylation enzyme activity reaction. Samples were subsequently separated by SDS–PAGE and digested in-gel with trypsin. Peptides were dried to completion and enriched with *O*-GlcNAc antibody (PTM-Biolabs Co., Ltd, PTM20160616C0) and analyzed by tandem mass spectrometry (MS/MS) in Q Exactive (Thermo).

**Expanded View** for this article is available online.

### Acknowledgements

We thank Dr. Yingfang Liu (Institute of Biophysics, Chinese Academy of Sciences) for making the structural model of SEC, and the *Arabidopsis* Biological Resource Center for the seeds of SALK_106339 and SALK_034290. This work was funded by the Basic Science Center Project of National Natural Science Foundation of China (31788103) and the National Key Research and Development Program of China (2016YFD0101004).

### Author contributions

LX designed experiments and performed immunoblotting, phenotyping, and data analysis. YL performed ChIP, immunoblotting, qRT–PCR, and plant material preparation. SX performed qRT–PCR and transgenic analyses. JX provided useful critiques of the manuscript. BW performed plasmid construction. HD

prepared plant materials. ZL analyzed mass-spectrum data. YX analyzed data. KC designed experiments, KC and LX wrote the manuscript. All authors commented on the manuscript.

## Conflict of interest

The authors declare that they have no conflict of interest.

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
