## [Review Process File · The EMBO Journal]

***Arabidopsis* O-GlcNAc transferase SEC activates histone methyltransferase ATX1 to regulate flowering**

Lijing Xing, Yan Liu, Shujuan Xu, Jun Xiao, Bo Wang, Hanwen Deng, Zhuang Lu, Yunyuan Xu, Kang Chong

Review timeline:	Submission date:	30th Aug 17
	Editorial Decision:	4th Oct 17
	Revision received:	16th Mar 18
	Editorial Decision:	23rd Apr 18
	Revision received:	21st Jul 18
	Accepted:	30th Jul 18

Editors: Ieva Gailite / Deniz Senyilmaz Tiebe

Transaction Report:

1st Editorial Decision

4th Oct 17

Thank you for submitting your manuscript for consideration by The EMBO Journal. We have now received two referee reports on your manuscript, which are included below for your information. In light of these comments, we unfortunately had to conclude that the study is not a sufficiently strong candidate for publication in The EMBO Journal.

As you can see, while referee #1 appreciates the identification of ATX1 O-GlcNAcylation in flowering time regulation, referee #2 points out the limited in vivo relevance of the proposed mechanism, and indicates a number of crucial experimental issues. Although referee #1 is more positive in their evaluation, they also indicate similar concerns and point out that a number of essential further experiments are needed to convincingly substantiate the proposed pathway of flowering time regulation. Given these opinions from trusted experts in the field, I am afraid we cannot offer further proceedings towards publication in The EMBO Journal.

Thank you in any case for the opportunity to consider this manuscript. I am sorry that I cannot communicate more positive news, but nevertheless hope that you will find our referees' comments helpful.

REFeree COMMENTS

Referee #1:

In this manuscript Lijing Xing et al. describe the control of flowering time regulation through O-Glc-N-acylation. The manuscript is well written and the topic is interesting and highly timely. The presented work provides one of the first examples of O-Glc-N-acylation as a regulatory mechanism

of plant development. In a tour de force involving detailed biochemical analysis and genetic and transgenic plant analysis, the authors show that a mutant in one of the two Arabidopsis O-Glc-N-Ac-transferases, SEC, is early flowering and that this is because of hyper-repression of the floral repressor FLC. They go on to show that SEC activates the H3K4 histone methyltransferase ATX1 through O-Glc-N-Acylation and that the loss of this modification results in reduced activity of ATX1 and in turn in reduced histone H3K4 trimethylation at the FLC locus. Although the effect is relatively small, the presented detailed biochemical work makes a convincing case for the proposed mechanism. The conclusions are mostly well supported and the manuscript should be interesting to a wide range of chromatin biologists and plant biologists.

There are some points where the manuscript could be improved:

- 1) The authors use the novel allele *sec-5* but include previously described alleles in Fig 1A. Has the flowering phenotype been previously noted in *sec-1*, *sec-2*, *sec-3*?
- 2) Fig 2: The mode of normalization of the ChIP is not clear. Please state in the legend how it was normalized (to input only?, to H3? both?). It appears that each amplicon was normalized to the respective Col value, which removes information on which amplicon has the highest enrichment, and thus where along the locus the peak of the enrichment is located. Please supply this information.
- 3) Fig 3A: The Yeast-Two-Hybrid experiment is missing controls. Please show the result of the following combinations BD-ATX1/empty AD and empty BD/AD-Sec and empty BD/AD-SEC-N, which are vital controls.
- 4) Fig 3B: The CTD110.6 immunoblot is not very convincing. It looks like the reduced signal on the left is caused by a smudge on the blot. Please supply a more convincing image.
- 5) Fig 6C, F: The difference in flowering time is very small and it seems the wt construct does not fully complement the *atx1* mutant (although this is difficult to judge because not Col is included in panel C). Are these differences statistically significant?
- 6) Fig 1C, F, Fig 6, Fig S1 and others throughout ms: The label "Relative enrichment" on the y-axis for expression data is not appropriate. Replace with "Relative transcript levels".
- 7) Discussion, last paragraph: Please clarify the relationship between the HMT ATX1 and COMPASS. It is not clear whether ATX1 is the catalytic subunit of COMPASS or not. Please provide a reference for the statement that ATX1 is required for recruitment of AtCOMPASS-like to promoters.
- 8) Legend Fig 3D: The statement on the n of the flowering time experiment is duplicated. Which of the two is correct?

Referee #2:

The O-GlcNac modification has been found on thousands of proteins that regulate diverse cellular pathways. In Arabidopsis, O-GlcNac is predicted to be applied by two enzymes, Secret Agent (SEC) and SPINDLY (SPY). In this manuscript, the authors investigated the effect of O-GlcNac on flowering time regulation. The authors found that loss of SEC function causes early flowering associated with reduced expression of FLC. They found reduced H3K4me3 present on FLC, which they correlate with reduced ATX1 activity. They found that ATX1 is modified O-GlcNac and identified the serine residue likely containing this modification. Nevertheless, mutating this residue did not have a significant effect on flowering time regulation, questioning the relevance of O-GlcNac on ATX1 activity. Together, while the authors provide evidence that ATX1 is modified by O-GlcNac, like thousands of other proteins, the relevance of this modification for ATX1 activity and flowering time regulation seems questionable.

Major points:

- The authors claim that *sec-5* mutant is early flowering; however, this phenotype is not described in detail. The authors should clearly show how the *sec-5* early flowering phenotype compares to wild type and other early flowering mutants, for instance *atx1-2* or *flc* alleles.
- To support their claims the authors should provide flowering time data for a *sec-5 atx1-1* double mutant.
- The majority of flowering time experiments lack statistical treatments (e.g. Figure 3C, Figure 6C,F, Figure S2A) and the differences between genotypes are minor and probably not biologically relevant.
- To evaluate the data of Figure 2, it would be important to know the age of the analyzed plants and the level of FLC expression. Furthermore, it is unclear how the data shown in panel 2f were normalized. To use the *sec* mutant as control is not appropriate, they should show a locus not modified by SEC. Lastly, the order of the columns should be the same as in the other figure panels

of Figure 2.

-Figure 3B: This figure is not really convincing. There is a background signal also in the sample not expressing the SEC enzyme.

-Figure 4D: There is less ATX1 in the *sec-5* mutant compared to the wild-type control, which may explain the depletion of H3K4me3 observed in the *sec-5* mutant on this blot. Given the subtle differences it would be important to know how many times this experiment was repeated and similar results were obtained.

Minor points:

-The reference Lazarus MB et al. Structure of human O-GlcNAc transferase and its complex with peptide substrate. Nature 2011 appears twice in the reference list.

-Legend figure 3D: The indication how many plants were used appears twice in the legend, but the numbers differ: "more than 20 plants...", "more than 15 plants..."

-Full names for proteins should be shown in the introduction: PIF3, JAZ1, MAF1, SOC1, VER2.

-The y axis of graphs showing expression data are labelled with "relative enrichment". The correct labelling should be "relative expression". It should be clearly outlined how data were normalized to come to relative expression values

1st Revision - authors' response

16th Mar 18

Thank you very much for your critical comments and constructive suggestions on our manuscript which was previously submitted with the title of "O-GlcNAc transferase SEC activates ATX1 to exert epigenetic flowering regulation in *Arabidopsis*". We have addressed the comments of two referees with new data involved in the new figures and revised our original manuscript version.

Two aspects of this manuscript encourage us to submit it to the top journal. Especially, our finding is a direct linkage between histone methylation and protein O-GlcNAc modification, which is a novel modification to regulate histone methylation in epigenetic regulation in plants.

1. O-linked b-D-N-acetylglucosamine (O-GlcNAc) modification is one of the key points to regulate a series of biological process in cells. However, little is known about the O-GlcNAc modification of histone lysine methyltransferase (HKMT) in developmental process. Here, we report that O-GlcNAc transferase in *Arabidopsis*, SECRET AGENT (SEC), directly activates trithorax histone lysine methyltransferase (HKMT) ATX1 via modifying it with O-GlcNAc to regulate flowering. Ser 947 in ATX1 SET domain is necessary for O-GlcNAcylation-mediated activation. This topic is attractive for the readers in a wide research field, not only for plant science but also for animal even human science.
2. Epigenetic-based flowering regulation is well known in *Arabidopsis*. Histone methylation is one of the key types to regulate plant development. However, whether O-GlcNAcylation plays a role in epigenetic regulation in plant? We show that loss of function of *SEC* inhibited ATX1-regulated H3K4me3 level of *FLC*, a key negative regulator of flowering, leading to weak early flowering phenotype in *Arabidopsis*. These findings indicate the role of O-GlcNAc signaling in epigenetic regulation during plant development, and shed light on crosstalk between histone methylation-related epigenetic process and O-GlcNAc signaling. It is a novel finding in plant biology.

Response to Referee #1:

In this manuscript Lijing Xing et al. describe the control of flowering time regulation through O-Glc-N-acylation. The manuscript is well written and the topic is interesting and highly timely. The presented work provides one of the first examples of O-Glc-N-acylation as a regulatory mechanism of plant development. In a tour de force involving detailed biochemical analysis and genetic and transgenic plant analysis, the authors show that a mutant in one of the two *Arabidopsis* O-Glc-N-Ac-transferases, SEC, is early flowering and that this is because of hyper-repression of the floral repressor FLC. They go on to show that SEC activates the H3K4 histone methyltransferase ATX1 through O-Glc-N-Acylation and that the loss of this modification results in reduced activity of ATX1 and in turn in reduced histone H3K4 trimethylation at the *FLC* locus. Although the effect is relatively small, the presented detailed biochemical work makes a convincing case for the proposed mechanism. The conclusions are mostly well supported and the manuscript should be interesting to a

wide range of chromatin biologists and plant biologists.

There are some points where the manuscript could be improved:

- 1) The authors use the novel allele *sec-5* but include previously described alleles in Fig 1A. Has the flowering phenotype been previously noted in *sec-1*, *sec-2*, *sec-3*?

Response: We added the flowering phenotype description to the first part of "Results" in the revised version.

- 2) Fig 2: The mode of normalization of the ChIP is not clear. Please state in the legend how it was normalized (to input only?, to H3? both?). It appears that each amplicon was normalized to the respective Col value, which removes information on which amplicon has the highest enrichment, and thus where along the locus the peak of the enrichment is located. Please supply this information.

Response: In the revised version, we analyzed the ChIP results by normalizing the data to input. The normalization of ChIP is added to the Fig2 legend in the revised version.

- 3) Fig 3A: The Yeast-Two-Hybrid experiment is missing controls. Please show the result of the following combinations BD-ATX1/empty AD and empty BD/AD-Sec and empty BD/AD-SEC-N, which are vital controls.

Response:

Response: We repeated the Yeast-Two-Hybrid experiment and the combinations of BD-ATX1/ empty AD and empty BD/AD-SEC and empty BD/AD-SEC-N were added to Fig3A.

- 4) Fig 3B: The CTD110.6 immunoblot is not very convincing. It looks like the reduced signal on the left is caused by a smudge on the blot. Please supply a more convincing image.

Response: We repeated the experiment of Fig3B, and new data was used in Fig3B in the revised version.

- 5) Fig 6C, F: The difference in flowering time is very small and it seems the wt construct does not fully complement the *atx1* mutant (although this is difficult to judge because not Col is included in panel C). Are these differences statistically significant?

Response: We repeated this biological experiment and compared the flowering phenotype Col-0, atx1-2, atx1-2 35S::ATX1-FLAG and atx1-2 35S::ATX1m-FLAG. Together with the real-time qPCR results, our new data showed that WT construct complemented the atx1-2 mutant. In the revised version, in addition to rosette leaf numbers, we further analyzed the visible buds at day 21 and bolting rate at day 24 after germination, the results showed that there are significant differences between WT and atx1-2 35S::ATX1m-FLAG lines, indicating the ATX1 function in flowering regulation is significantly inhibited in the case of Ser947 mutation.

- 6) Fig 1C, F, Fig 6, Fig S1 and others throughout ms: The label "Relative enrichment" on the y-axis for expression data is not appropriate. Replace with "Relative transcript levels".

Response: The description of "Relative enrichment" in Fig 1C, F, Fig 6, Fig S1 is corrected in the revised version.

- 7) Discussion, last paragraph: Please clarify the relationship between the HMT ATX1 and COMPASS. It is not clear whether ATX1 is the catalytic subunit of COMPASS or not. Please provide a reference for the statement that ATX1 is required for recruitment of AtCOMPASS-like to promoters.

Response: We added two references in the discussion part, indicating that ATX1 is a subunit of COMPASS complex. The reference of Ding et al (2012) is for the statement that ATX1 is required for recruitment of AtCOMPASS-like to promoters.

- 8) Legend Fig 3D: The statement on the n of the flowering time experiment is duplicated. Which of the two is correct?

Response: We repeated the experiment and statistical analysis was done again, the data of statistical analysis was used in the revised version.

Response to Referee #2

The *O*-GlcNac modification has been found on thousands of proteins that regulate diverse cellular pathways. In Arabidopsis, *O*-GlcNac is predicted to be applied by two enzymes, Secret Agent (SEC) and SPINDLY (SPY). In this manuscript, the authors investigated the effect of *O*-GlcNac on flowering time regulation. The authors found that loss of SEC function causes early flowering associated with reduced expression of *FLC*. They found reduced H3K4me3 present on *FLC*, which they correlate with reduced ATX1 activity. They found that ATX1 is modified *O*-GlcNac and identified the serine residue likely containing this modification. Nevertheless, mutating this residue did not have a significant effect on flowering time regulation, questioning the relevance of *O*-GlcNac on ATX1 activity. Together, while the authors provide evidence that ATX1 is modified by *O*-GlcNac, like thousands of other proteins, the relevance of this modification for ATX1 activity and flowering time regulation seems questionable.

Response: In the revised version, we added further flowering phenotype analysis, and our data showed that S947A indeed reduced ATX1 function in regulating flowering time. The 35S::ATX1-FLAG can rescue the flowering phenotype of atx1-2. Nevertheless, atx1-2 35S::ATX1m-FLAG plants exhibited early flowering phenotype comparing with Col-0 (Figure 6A, 6B), showing significant difference at levels of visible buds, bolting rates and rosette leaf numbers (Figure 6C-6E). In addition to the role of Ser947 O-GlcNAcylation in regulating ATX1 activity, it is likely that there are other mechanisms for ATX1 activity regulation.

Major points:

-The authors claim that *sec-5* mutant is early flowering; however, this phenotype is not described in detail. The authors should clearly show how the *sec-5* early flowering phenotype compares to wild type and other early flowering mutants, for instance *atx1-2* or *flc* alleles.

-To support their claims the authors should provide flowering time data for a *sec-5 atx1-1* double mutant.

Response: Actually, we have repeated the phenotype experiments so many times. The early flowering phenotype of sec-5 is reproducible every time. According to the comments, we repeated the flowering phenotype again and did detailed statistical analysis again. In the revised version, in addition to rosette leaf numbers, we further analyzed the visible buds at day 21 and bolting rate at day 24 after germination. The data showed significant difference between sec-5 and WT. We have generated sec-5 atx1-2 double mutant before, we added detailed flowering phenotype analysis among Col, sec-5, atx1-2, sec-5 atx1-2 and flc-3 in the revised version, these data were shown in Table1, FigureS4, TableS1 and TableS2.

(1) The majority of flowering time experiments lack statistical treatments (e.g. Figure 3C, Figure 6C, F, Figure S2A) and the differences between genotypes are minor and probably not biologically relevant.

Response: In the revised version, we added statistical analysis of flowering phenotype; moreover, more plants were used for further statistical analysis. We compared the early flowering phenotype of sec-5 with reported flc-3, sec-5 showed similar early flowering phenotype with flc-3, indicating the biological relevance of phenotype observation. Generally, the phenotype of early flowering mutants are weaker than that of late-flowering mutants. And early flowering mutants show stronger phenotype under SD condition. In our manuscript, sec-5 shows stronger phenotype under SD (Figure 1). Based on the statistical analysis of rosette leaf numbers, days to first flower open, visible buds and bolting rate, the flowering phenotypes are of biological significance.

Reference:

Wen-Hui Shen et al., 2015, The Plant Journal, 81, 316–328.

Ste'phane Pien et al., 2008, Plant Cell, Vol. 20: 580–588.

(2) To evaluate the data of Figure 2, it would be important to know the age of the analyzed plants and the level of *FLC* expression. Furthermore, it is unclear how the data shown in panel 2f were normalized. To use the *sec* mutant as control is not appropriate, they should show a locus not

modified by SEC. Lastly, the order of the columns should be the same as in the other figure panels of Figure 2.

Response: The age plants for ChIP analysis is 12-d-old, and we added this data to the Figure legend of Fig2. The data was normalized to input, which was added to the figure legend. And we deleted the data of SEC enrichment on FLC chromatin in the revised version.

(3) Figure 3B: This figure is not really convincing. There is a background signal also in the sample not expressing the SEC enzyme.

Response: We repeated the experiment of Figure3B and the new data was used in the revised version.

(4) Figure 4D: There is less ATX1 in the *sec-5* mutant compared to the wild-type control, which may explain the depletion of H3K4me3 observed in the *sec-5* mutant on this blot. Given the subtle differences it would be important to know how many times this experiment was repeated and similar results were obtained.

Response: In the revised version, the activity of ATX1 in Col-0 and sec-5 was normalized by H3 and input, indicating that loss of SEC function reduced HKMT activity of ATX1. Based on the data of Figure 4C, we further analyzed the protein level of ATX1 in sec-5 comparing with Col-0 at different development stages, the repeated data showed that the protein expression of ATX1 is decreased in sec-5 at the development stages of before and after flowering (Figure 4E). This experiment was repeated three times. Combined with the activity analysis data in vitro and in vivo, the data indicated that SEC play roles not only in activating ATX1 but also regulating ATX1 protein stability.

Minor points:

(1) The reference Lazarus MB et al. Structure of human *O*-GlcNac transferase and its complex with peptide substrate. Nature 2011 appears twice in the reference list.

Response: We checked carefully and revised the reference list.

(2) Legend figure 3D: The indication how many plants were used appears twice in the legend, but the numbers differ: "more than 20 plants...", "more than 15 plants..."

Response: We analyzed the flowering phenotype with more plants and new statistical data were used in the revised version.

(3) Full names for proteins should be shown in the introduction: PIF3, JAZ1, MAF1, SOC1, VER2.

Response: The full names of PIF3, JAZ1, MAF1, SOC1 and VER2 were used in the revised version.

(4) The y axis of graphs showing expression data are labelled with "relative enrichment". The correct labelling should be "relative expression". It should be clearly outlined how data were normalized to come to relative expression values.

Response: The description of "relative enrichment" was corrected, and the normalization of data was added in the revised version.

Thank you for submitting a revised version of your manuscript. It has now been seen by both of the original referees whose comments are shown below.

As you will see they both find that all criticisms have been sufficiently addressed and recommend the manuscript for publication. However, before we can officially accept the manuscript there are a few editorial issues concerning text and figures that I need you to address.

Please address the remaining points that are raised by the referees.

 REFEREE COMMENTS

Referee #1:

The revised version of the manuscript has much improved and my previous concerns have largely been addressed. However, a few points remain to be clarified before publication can be recommended.

ad 1) Hartweck et al 2006 J Exp Bot report that *sec-1* (Ws background) neither has a flowering phenotype in long nor short days, but it has altered leaf initiation rates. This may be an ecotype difference and should be discussed clearly (as such). However, since only one of the reported alleles appears to be late flowering, more emphasis is to be placed on transgenic complementation. While FLC levels are restored in *sec-5 SEC::SEC* to wild type levels, this is not true for rosette leaf number. Thus, there remains a possibility that a second site mutation is responsible for the flowering aka rosette leaf number phenotype. Please clarify.

The manuscript contains some typos and grammar mistakes and would benefit from careful language editing.

Referee #2:

The authors made efforts to address my concerns and improved the manuscript. I have some remaining concerns that require to be addressed:

Figure 6: Difference in FLC expression between *atx, 35S::ATX* and *atx, 35S::ATXm* are marginal and only the results of one experiment are shown. The authors should show the data of the other two replicates in the supplement to allow evaluating whether the differences are indeed significant.

Figure 2: Data of one representative replicate are shown. The data of the other replicates should be shown in the supplement.

Figure 3E-G: The correct test that should be done is to compare whether phenotypes observed in *sec-5* significantly differ from those in *sec-5; 35S::ATX1* to support the statement that "*sec-5 35S::ATX1-HA* transgenic plants showed similar rosette leaf numbers as that of *sec-5*".

Figure S2C: The authors claim that the early flowering phenotype of *sec5* is complemented by expressing *SEC5::SEC*. However, based on the data shown in Figure S2C, the leaf numbers at flowering are not different in complementing lines compared to the mutant. I assume the statistical difference that has been indicated relates to differences to wild type; therefore, the complemented lines remain as early flowering as the mutant. This requires an explanation.

Minor point:

In the legend of Table 1, it should be written that phenotypes were scored at 21 or 24 days after germination, not "in seeds germinated for 21/24 days".

 2nd Revision - authors' response

21st Jul 18

Thank you very much for considering our manuscript to be accepted by EMBO J. We are submitting the revision version now. We have addressed all the comments of two reviewers, and our response to the referees and editors are attached at the end of the letter.

The changes to our manuscript are summarized as follows:

1. According to the comments of two referees on previous Fig S2C, we repeated the flowering phenotype analysis with increased sample number, and added the assay of days to first flower opening. The new data showed that the rosette leaf number and days to first flowering opening were restored in *sec-5 pSEC::SEC* plants. At the same time, we have generated *SEC* overexpressing transgenic lines in Col-0 background which exhibited late flowering phenotype with increased rosette leaf numbers. These new data support that loss of *SEC* function reduced

FLC expression and rosette leaf number, and days to flowering. Therefore, the previous FigS2 was changed to Fig EV1E-H in the revised version.

2. The previous Fig2 showing the data of one experiment was changed to new Fig2 to show the mean of three independent biological replicates.
3. The previous Fig 6B showed the data of one experiment, which was replaced with the mean of three independent biological replicates in the revised version.
4. The colors of bars in Fig6 B-E were altered in the revised version. And the order of Fig 6C, D and E was exchanged, the previous E was shown as C, C as D and D as E in the revised version.
5. The revised or added sentences, and EV and appendix figure callouts are marked with red color.

Response to referees' comments

Referee #1:

The revised version of the manuscript has much improved and my previous concerns have largely been addressed. However, a few points remain to be clarified before publication can be recommended.

ad 1) Hartweck et al 2006 J Exp Bot report that *sec-1* (Ws background) neither has a flowering phenotype in long nor short days, but it has altered leaf initiation rates. This may be an ecotype difference and should be discussed clearly (as such). However, since only one of the reported alleles appears to be late flowering, more emphasis is to be placed on transgenic complementation. While *FLC* levels are restored in *sec-5 SEC::SEC* to wild type levels, this is not true for rosette leaf number. Thus, there remains a possibility that a second site mutation is responsible for the flowering aka rosette leaf number phenotype. Please clarify.

Response:

Thanks for the constructive comments. We have discussed the development and leaf initiation rate phenotypes clearly in the first result part in the revised version. The previous rosette leaf number analysis was based on a limited sample size with a total number of less than 15 plants scored for statistical analysis. To further confirm the flowering time phenotype, we have repeated with three more biological experiments in which a total number of 52 plants were scored for rosette leaf number analysis, and the new data showed that the rosette leaf number was restored in *sec-5 pSEC::SEC* plants. At the same time, we also analyzed the flowering time indicated by "days to first flower opening", showing that the flowering time was restored in *sec-5 pSEC::SEC* plants. On the other hand, we have generated *SEC* overexpressing transgenic lines in Col-0 background which exhibited late flowering phenotype with increased rosette leaf numbers (see the figure below). These new data support that loss of *SEC* function reduced *FLC* expression and rosette leaf number, and days to flowering. Therefore, the previous FigS2 was changed to Fig EV1E-H in the revised version.

Figures for Referees removed.

2) The manuscript contains some typos and grammar mistakes and would benefit from careful language editing.

Response: Thanks for the comment. We have done the language editing according to the reviewer's suggestion.

Referee #2

The authors made efforts to address my concerns and improved the manuscript. I have some remaining concerns that require to be addressed:

Figure 6: Difference in *FLC* expression between *atx, 35S::ATX* and *atx, 35S::ATXm* are marginal and only the results of one experiment are shown. The authors should show the data of the other two replicates in the supplement to allow evaluating whether the differences are indeed significant.

Response: Thanks for the comment. In the revised version, the previous data in Fig6 B which only showed one replicate was changed to the mean of three biological replicates, and the data of all three biological replicates were shown as Appendix FigS3. The colors of bars in Fig6 B-E were altered in the revised version. And the order of Fig 6C, D and E was exchanged, the previous E was shown as C, C as D and D as E in the revised version.

Figure 2: Data of one representative replicate are shown. The data of the other replicates should be shown in the supplement.

Response: Thanks for the comment. The previous data in Fig2 which only showed one replicate was changed to the mean of three biological replicates, and all three biological replicates data were shown as Appendix FigS1 in the revised version.

Figure 3E-G: The correct test that should be done is to compare whether phenotypes observed in *sec-5* significantly differ from those in *sec-5; 35S::ATX1* to support the statement that "*sec-5 35S::ATX1*-HA transgenic plants showed similar rosette leaf numbers as that of *sec-5*".

Response: Thanks for the constructive comment. We have done the test to compare the rosette leaf number, and visible buds at days 21 and bolting rate at days 24 after germination between *sec-5* and *sec-5 35S::ATX1* lines. The differences of leaf numbers and percent of visible buds and bolting rates between *sec-5* and *sec-5 35S::ATX1* lines are not significant. The tests are shown in Fig 3E in the revised version.

Figure S2C: The authors claim that the early flowering phenotype of *sec5* is complemented by expressing *SEC5::SEC*. However, based on the data shown in Figure S2C, the leaf numbers at flowering are not different in complementing lines compared to the mutant. I assume the statistical difference that has been indicated relates to differences to wild type; therefore, the complemented lines remain as early flowering as the mutant. This requires an explanation.

Response:

Thanks for the constructive comments. We have discussed the development and leaf initiation rate phenotypes clearly in the revised version. The previous rosette leaf number analysis was based on a limited sample size with a total number of less than 15 plants scored for statistical analysis. To further confirm the flowering time phenotype, we have repeated with three more biological experiments in which a total number of 52 plants were scored for rosette leaf number analysis, and the new data showed that the rosette leaf number was restored in *sec-5 pSEC::SEC* plants. At the same time, we also analyzed the flowering time indicated by "days to first flower opening", showing that the flowering time was restored in *sec-5 pSEC::SEC* plants. On the other hand, we have generated *SEC* overexpressing transgenic lines in Col-0 background which exhibit late flowering phenotype with increased rosette leaf numbers (see the figure below). These new data support that loss of *SEC* function reduced *FLC* expression and rosette leaf number, and days to flowering. Therefore, the previous FigS2 was changed to Fig EV1E-H in the revised version.

Figures for Referees removed.

Minor point:

In the legend of Table 1, it should be written that phenotypes were scored at 21 or 24 days after germination, not "in seeds germinated for 21/24 days".

Response: We have rewritten the description of the phenotype in the revised version.

3rd Editorial Decision

30th Jul 18

Thank you for submitting the revised version of your manuscript. I have now looked at everything and all looks fine. Therefore, I am very pleased to inform you that your manuscript has been accepted for publication at The EMBO Journal.

Congratulations on the very nice work!

Corresponding Author Name: Kang Chong

Journal Submitted to: EMBO J

Manuscript Number: EMBOJ-2017-98115R-Q